# Integrin Signaling Shaping BTK-Inhibitor Resistance

**DOI:** 10.3390/cells11142235

**Published:** 2022-07-18

**Authors:** Laura Polcik, Svenja Dannewitz Prosseda, Federico Pozzo, Antonella Zucchetto, Valter Gattei, Tanja Nicole Hartmann

**Affiliations:** 1Department of Internal Medicine I, Faculty of Medicine and Medical Center, University of Freiburg, 79106 Freiburg, Germany; laura.polcik@uniklinik-freiburg.de (L.P.); svenja.dannewitz@uniklinik-freiburg.de (S.D.P.); 2Faculty of Biology, University of Freiburg, 79106 Freiburg, Germany; 3Clinical and Experimental Onco-Hematology Unit, Centro di Riferimento Oncologico di Aviano (CRO) IRCCS, 33081 Aviano, Italy; federico.pozzo@cro.it (F.P.); zucchetto.soecs@cro.it (A.Z.); vgattei@cro.it (V.G.)

**Keywords:** CLL, VLA-4, CD49d, BTK, therapy resistance, ibrutinib, acalabrutinib, zanubrutinib, pirtobrutinib, animal models

## Abstract

Integrins are adhesion molecules that function as anchors in retaining tumor cells in supportive tissues and facilitating metastasis. Beta1 integrins are known to contribute to cell adhesion-mediated drug resistance in cancer. Very late antigen-4 (VLA-4), a CD49d/CD29 heterodimer, is a beta1 integrin implicated in therapy resistance in both solid tumors and haematological malignancies such as chronic lymphocytic leukemia (CLL). A complex inside-out signaling mechanism activates VLA-4, which might include several therapeutic targets for CLL. Treatment regimens for this disease have recently shifted towards novel agents targeting BCR signaling. Bruton’s tyrosine kinase (BTK) is a component of B cell receptor signaling and BTK inhibitors such as ibrutinib are highly successful; however, their limitations include indefinite drug administration, the development of therapy resistance, and toxicities. VLA-4 might be activated independently of BTK, resulting in an ongoing interaction of CD49d-expressing leukemic cells with their surrounding tissue, which may reduce the success of therapy with BTK inhibitors and increases the need for alternative therapies. In this context, we discuss the inside-out signaling cascade culminating in VLA-4 activation, consider the advantages and disadvantages of BTK inhibitors in CLL and elucidate the mechanisms behind cell adhesion-mediated drug resistance.

## 1. Introduction

Chronic lymphocytic leukemia (CLL) is a B cell neoplasm that belongs to the indolent non-Hodgkin lymphomas. In recent years, new agents have led to a paradigm shift in CLL patient care from chemo(immuno)therapy to targeted therapy, with unprecedented efficacy of the novel small molecule inhibitors of the B cell receptor (BCR) signaling or apoptosis pathways. Among the various new targeted therapy options, covalent Bruton’s tyrosine kinase (BTK) inhibitors such as ibrutinib were the pioneers of success in this disease [1]. Despite enormous achievements, toxicity and drug resistance in a part of the patients, which may also be due to the need for continuous use of the drug, represent issues of these inhibitors. The development of resistance can also be caused by minimal residual disease upon therapy that escapes the targeted agents. The surviving tumor cells are located in tissue sites such as the bone marrow and the secondary lymphoid organs. Only in these environments, in close contact with antigen and/or immune or other accessory cells such as T lymphocytes and stromal cells, can CLL cells proliferate despite therapy [2]. The positioning of CLL cells in these organs is mediated by adhesion molecules such as integrins.

The integrin very late antigen-4 (VLA-4), also known as CD49d, is a strong and reliable predictor of poor outcomes in CLL and has a well-studied biological role. VLA-4 ligands are expressed on follicular dendritic cells (FDCs) in lymphoid organs and strengthen the antigen synapse during BCR stimulation while at the same time assuring the retention and positioning of the malignant cells in this microenvironment (reviewed in Ref. [3]). BTK is a key component of BCR signaling but also acts downstream of other cell surface receptors, such as adhesion molecules, chemokine receptors and toll-like receptors (TLRs) [4,5]. It is therefore not surprising that BTK inhibitors such as ibrutinib disrupt tumor cell-microenvironment interactions and mobilize tumor cells from the tissue sites into the peripheral blood, manifesting in an initial increase in lymphocytosis at therapy start [6]; however, patients expressing high levels of CD49d exhibit a lower degree of lymphocytosis under BTK inhibitor therapy, along with worse progression-free survival (PFS), compared to patients with low CD49d expression. This suggests a role of this integrin and/or its signaling in resistance development [7], and this review explains the mechanistic role of integrin function in drug resistance and dissects specific aspects of the signaling cascade that trigger integrin activation. Furthermore, we discuss how the VLA-4 integrin and the BCR cooperate and how signaling components between these molecules could bypass BTK inhibition. We describe how we could learn from animal models to overcome the resistance and finally elude alternative inhibitors in this perspective. 

## 2. Cell Adhesion-Mediated Drug Resistance

A plentitude of molecular mechanisms can contribute to primary and acquired drug resistance of tumor cells. Since the 1990s it has been known that the adhesion of cancer cells to the extracellular matrix or to stromal cells supports their survival during radiotherapy or chemotherapy, a phenomenon called cell adhesion-mediated drug resistance [8]. Particularly the adhesive crosstalk of tumor cells with stromal cells and its consequences on tumorigenesis, metastasis, and the development of resistance has been studied in detail. Nowadays, in view of the increasing knowledge about the role of the immune system in tumorigenesis, the concept can be further extended and differentiated. Cell-cell contacts of tumor cells and T cells or myeloid cells lead to the re-education of the immune system and thus, contribute to tumor immune evasion. Adhesion molecules are involved in this process by providing mechanical stability to the cellular contacts and function as an anchor by retaining cells in the tissue [9].

Integrins are heterodimeric adhesion molecules that are unique in their capability of transmitting signals from the microenvironment in intracellular cascades and back via bidirectional signaling. In humans, at least 24 heterodimers are known, which are comprised of non-covalently bound alpha- and beta-subunits, where the alpha-subunit often defines the specificity of the integrin to its ligand while the beta-subunit provides the anchoring to the cytoskeleton [10]. Integrins containing the beta1 (also known as CD29) subunit constitute the largest subgroup due to their numerous alpha subunit partners; this group has been described for its implication in several hallmarks of cancer and is the front runner for the development of drug or radiotherapy resistance as reviewed in [11].

Beta1 integrins mediate cell adhesion to the extracellular matrix by binding to e.g., collagen or fibronectin, or to immune or stromal cells that express integrin ligands such as vascular cell adhesion molecule 1 (VCAM-1). Upon engagement of their ligands, integrin clustering occurs, and focal contacts are built. The elaborate integrin adhesome near the intracellular beta domain and the associated signaling network trigger pathways involved in tumor resistance. Relevant components are e.g., focal adhesion kinase, Src, integrin-linked kinase, and phosphoinositol-3-kinase (PI3K). Further downstream, mitogen-activated protein kinases (MAPK), MYC, and nuclear factor kappa-light-chain enhancer of activated B cells (NF-κB) signaling is involved, pathways that can also be targeted for cancer therapy [12,13]. Components of the MAPK signaling pathway include p38 and p42/44 (also known as extracellular signal-regulated kinase, ERK), both of which mediate cell adhesion-dependent drug resistance [14]. In human melanoma, upregulation of fibronectin and beta1 integrins is able to overcome p38 inhibition and increase cell survival [15]. In leukemia, beta1-mediated adhesion to fibronectin downregulates pro-apoptotic Bim, whose activation is regulated by ERK and was linked to cell adhesion-mediated drug resistance [16,17,18]. In chronic myeloid leukemia, tumor cell adhesion to stromal cells induces ERK signaling and a chemoresistant phenotype [19]. MAPK signaling induces MYC transcription via degradation of the MYC antagonist mitotic arrest deficiency 1 (MAD1) [20]. MYC proteins are oncogenes that are implicated in 70% of cancers and are involved in promoting drug resistance in bladder, pancreatic and prostate cancer, but also in haematological malignancies such as multiple myeloma (MM) [21,22,23,24]. Beta1 integrins regulate MYC expression in mammary epithelial cells upon integrin engagement by fibronectin [25]; however, in human neuroblastoma cells MYC overexpression was shown to decrease beta1 integrin expression resulting in increased growth and tumorigenesis, pointing to complex MYC-beta1 integrin relations [26]. In drug-resistant breast cancer patients MYC is involved in the maintenance of chemotherapy-resistant cancer stem cells by increasing the generation of reactive oxygen species (ROS), metabolic byproducts of aerobic respiration that promote cancer cell growth and proliferation, and in turn, make cancer cells more susceptible to cytotoxicity resulting from additional ROS exposure [27,28]. ROS, on the other hand, can be generated by chemotherapeutics and radiation used in cancer treatment, and their production at sub-lethal levels promotes NF-κB activation [29,30]. NF-κB in turn suppresses cell death in cancer cells and was shown to confer drug resistance in human colon cancer cells by promoting the expression of the multidrug resistance gene 1 (*MDR1*) [31]. *MDR1* upregulation was linked to integrin αv-mediated NF-κB activation and cell adhesion-mediated drug resistance in glioblastoma [32]. Furthermore, NF-κB activation was shown to increase integrin beta1 expression in metastatic osteosarcoma and mediated radioresistance in human breast cancer cells, making the signaling pathway a possible alternative target to combat cell adhesion-mediated drug resistance [33,34].

## 3. VLA-4 Signaling—The Inside-Out Cascade

A key beta1 integrin member that is implicated in cell adhesion mediated radiotherapy or drug resistance of various solid and hematologic cancers, such as CLL (see later chapters) is the very late antigen-4 (VLA-4). The integrin consists of an alpha4 (CD49d) and a beta1 (CD29) subunit [3]. The major VLA-4 ligands are VCAM-1 and fibronectin [35,36]; however, ligands such as EMILIN-1 or osteopontin are also bound by VLA-4 [37,38,39]. In solid tumors, VCAM-1-VLA-4 interactions for example enhance metastasis of melanoma and breast cancer and reduce susceptibility to chemotherapy in ovarian cancer [40,41,42]. On the other hand, VLA-4 surface expression can be down-regulated during oncogenic transformation in breast cancer cells, presumably contributing to the invasive phenotype of these cells in an alpha5beta1 integrin-dependent manner [43]; this indicates that tumor progression is dependent on the dynamic up- and downregulation of VLA-4 on tumor cells depending on the state of the malignant cell and whether the disease progresses or is stagnant. In hematopoietic malignancies almost all tumor cell types express VLA-4 and the integrin is particularly well studied for its role in homeostasis and malignancy of hematopoietic stem and progenitor cells of the bone marrow microenvironment [44]. Moreover, in MM, a plasma cell disorder that is characterized by its osteolytic bone microenvironment, increased levels of VLA-4 are found on MM cells with active disease and associated to cell adhesion mediated drug resistance [8,45]. Targeting VLA-4 with micellar nanoparticles to inhibit adherence of MM cells to the protective microenvironment resulted in reduced tumor burden in a mouse model [46]. In Burkitt lymphoma cells, CD49d mediated cell adhesion-dependent drug resistance via NF-κB activation; this resistance can be overcome by treatment with the proteasome inhibitor bortezomib, which suppresses NF-κB activation [47].

Besides increased VLA-4 expression, a shift from a low- to a high-affinity state of the integrin might be even more critical to shaping the malignant disbalance in this disease. VLA-4, such as any integrin, can be triggered into this conformational state of strongest adhesive strength by a rapid intracellular signaling cascade; this cascade is called inside-out activation and is required to switch on the integrin and thus, precisely regulate its extent of functionality. Inside-out signaling starts with the activation of non-integrin cell surface receptors, e.g., cytokine-, chemokine-, or antigen receptors (Figure 1). 

For example, the cascade can start with the activation of CXCR4, CD44 or—in the case of B cells—BCR stimulation. Triggering of these surface receptors by their ligands (e.g., CXCL12, hyaluronic acid, or antigen) induces an intracellular signaling cascade involving PI3K and phospholipase C gamma (PLCγ) and various other kinases and G-protein coupled proteins [48,49]. In detail, Src tyrosine kinases, such as LYN, associate with the BCR and are able to directly phosphorylate and activate both, SYK and BTK [50]. SYK activation subsequently phosphorylates PI3K, which associates with the plasma membrane through docking sites such as CD19, where it phosphorylates phosphatidylinositol biphosphate (PIP2), resulting in PIP3, a second messenger. PIP3 then recruits effector proteins such as BTK, PLCγ and protein kinase C β (PKCβ). 

Another key inside-out component is the GTPase Ras-related protein 1 (Rap1) which can also be triggered by BCR clustering [51]. Rap1 belongs to the Ras superfamily of GTPases and can mediate VLA-4 signal propagation via CD31, a crucial integrin adhesion amplifier [52]. Furthermore, Rap1 binds the intracellular integrin adaptor talin-1 resulting in the activation of beta1 integrins [53]. Talin-1 is also known to be phosphorylated by PKC in an alternative, Rap1-independent pathway [54]. Talin-1 is crucial for beta1 integrin-mediated adhesion stabilization [55]. Similarly, kindlin-3 is essential for the integrin activation process [56]. In B cells, kindlin-3 is crucial for marginal zone B cell maintenance and puts the breaks on follicular B cell activation and differentiation [57]. Kindlin-3 can bind the cytoplasmic integrin adaptor paxillin, which regulates VLA-4-dependent adhesion strengthening by binding to the intracellular domain of the integrin [58]. Blocking the VLA-4-paxillin interaction was shown to impair integrin-dependent anchorage to the cytoskeleton and tethering [59]. Rap1 signaling can bypass the paxillin axis to induce integrin-mediated cell adhesion and spread in a focal adhesion kinase (FAK)-Src-signaling independent manner [60]. The inside-out signaling cascade and binding of integrin adaptors induce a series of rapid VLA-4 conformational changes resulting in a high-affinity state of the integrin that can be detected using FRET-based approaches and results in firm adhesion of the VLA-4-expressing cell to a substrate [61]. VLA-4 activation triggers Ca^2+^ mobilization which in turn is required to keep up integrin binding to extracellular matrix components [62,63].

It is controversially discussed whether BTK is part of inside-out integrin signaling. BTK belongs to the tec protein tyrosine kinase (TEC) family of non-receptor tyrosine kinases, which comprises five family members. BTK is primarily expressed in B cells and is found downstream of BCR signaling but can also act downstream of chemokine receptors or TLRs and enhance e.g., the production of inflammatory cytokines and interleukin 10 (IL-10) expression [4]. Besides being located downstream of BCR signaling components, BTK can occasionally be recruited up to the cell membrane through interaction with PIP_3_, where it plays a role in controlling antigen accumulation at the immune synapse [64]. Upon BCR activation the kinases SYK and LYN phosphorylate BTK at position Y551 in the kinase domain; this promotes BTKs catalytic activity and subsequently results in autophosphorylation at position Y223 in the Src homology (SH)_3_ domain [65,66,67]. Activated BTK induces downstream signaling through PLCγ2 and PKCβI which results in sustained Ca^2+^ influx necessary for adhesion and downstream MAPK signaling [68].

MAPK are not involved in inside-out integrin activation as integrin clustering takes place upstream of the MAPK cascade; however, upon engagement of the integrin with its ligand and stabilization of the adhesome, which is called outside-in signaling, MAPK are activated and can contribute to resistance signaling as described in the previous chapter [69].

## 4. BTK Inhibitor Therapy in Chronic Lymphocytic Leukemia

CLL is the most common type of leukemia in adults in Western countries and is characterized by the clonal expansion of malignant mature CD5^+^ B lymphocytes in the blood, bone marrow, and secondary lymphoid tissues [70]. Despite all recent success in CLL therapy, the disease is still not curable. Most of the novel drugs have to be taken continuously, suppressing the disease rather than eradicating it [71,72]; this issue can partly be attributed to the peculiarity of CLL cell proliferation: CLL cells expand only in interaction with surrounding cells within the primary and secondary lymphoid organs [3]. These form a reservoir for after therapy, from which clonal tumor evolution can take place, ultimately leading to the outgrowth of resistant tumors [73]. Within the lymphoid organs, CLL cells are supported by T lymphocytes, the myeloid compartment, and stromal cells, which interact to create a supportive cytokine environment for the tumor and induce survival signaling pathways such as the MAPK signaling cascade [74,75]. These pathways are also triggered by the direct cell-cell contact between the cells so that the cascades are constantly amplified. 

BTK inhibitors disrupt these amplification cycles by disrupting tumor cell-microenvironment interactions. For example, therapy with the covalent, orally bioavailable BTK inhibitor ibrutinib, one of the first targeted agents in CLL, leads to a rapid shrinkage of tumor mass in lymphoid organs, caused in part by the redistribution of CLL cells from the tissue into the bloodstream. In the circulation, the tumor cells lack supportive signals from accessory cells, which eventually leads to the death of the leukemic cells [7,76]. Redistribution manifests as transient lymphocytosis observed in 57% of first-line and 69% of relapsed/refractory CLL patients on ibrutinib treatment and resolves after an average of 14 weeks after treatment initiation. Prolonged lymphocytosis is associated with a favorable disease prognosis. In relapsed/refractory patients, the 5-year PFS rate with ibrutinib treatment constitutes 44%, and up to 92% in naïve patients [77,78,79,80].

Downsides of Ibrutinib include several adverse effects, namely diarrhea, respiratory tract infections, bleeding and cardiac side effects such as atrial fibrillation, and are likely caused by the off-target activity of the inhibitor which affects several other tyrosine kinase pathways besides BTK [81,82]. These side effects prompted ibrutinib therapy discontinuation in 41% of patients in a large US multicenter study [83]; however, grade 3 or 4 toxicities were infrequent in patients with relapsed/refractory CLL, including infections in 7.1%, anemia and neutropenia in 6% and 15% of patients, respectively [84]. Another issue is the need for continuous drug administration, which requires a high level of patient compliance, can reduce their quality of life, and is associated with high costs [85]. With daily drug administration, the risk of acquired resistance increases [86]. Resistance can be caused by point mutations in BTK, most importantly C481S/F/C/R, T474, L528 and T316A or gain-of-function mutations in PLCγ2, an enzyme required for BTK-mediated amplification of BCR signaling, but might also occur independent of mutational events due to microenvironmental interactions and bypass signaling pathways [87,88].

## 5. Next-Generation BTK Inhibitors

Resistance issues also arise with next-generation covalent BTK inhibitors. Acalabrutinib received FDA approval in 2019. [89] This second-generation inhibitor is successful in treating patients with relapsed CLL including high-risk patients [90], and may have fewer off-target toxicities than observed with ibrutinib due to higher target selectivity [91]. A phase III trial compared acalabrutinib to ibrutinib efficacy and safety in previously treated CLL patients and reported less treatment discontinuation due to adverse events in acalabrutinib- compared to ibrutinib-treated patients as well as similar PFS of 95% in both treatment groups [91]. Similar resistance-conferring mutations in BTK were discovered under acalabrutinib treatment to those observed under ibrutinib [92].

On 14th November 2019 the FDA granted accelerated approval to the second generation oral BTK inhibitor zanubrutinib for the treatment of mantle cell lymphoma, with several ongoing or concluded clinical studies for the treatment of CLL [93,94,95,96,97]. Zanubrutinib, same as the previous generation of BTK inhibitors, covalently binds the C481 residue of BTK and treatment with zanubrutinib resulted in less off-target events compared to ibrutinib [96,97]. One study including treatment naïve CLL patients with unfavorable prognosis due to del17p mutation reported an estimated PFS rate of 88.6% over 18-month treatment, with overall high tolerability and adverse effects reported in under 10% of patients. Furthermore, relapsed or refractory CLL patients without previous exposure to BTK inhibitor therapy had an ORR of 84.6% with an estimated PFS of 87.2% at 12.9 months follow-up [78]. Adverse effects in this study included grade 3 infections in 38.5% of participating patients, a higher rate compared to 24–30% for ibrutinib-treated patients in previous independent studies [78,97,98]. A recent phase 2 study reported continued disease control or improved response rates of ibrutinib- and/or acalabrutinib-intolerant relapsed/refractory CLL patients under zanubrutinib treatment, with a 94% disease control rate and overall fewer off-target binding compared to ibrutinib and acalabrutinib [99]. Another phase 3 study directly comparing zanubrutinib versus ibrutinib treatment in relapsed/refractory CLL is currently ongoing [93]. Moreover, previously untreated CLL patients were evaluated after zanubrutinib treatment in combination with obinutuzumab and the B cell lymphoma-2 (BCL-2) inhibitor venetoclax in a recently published phase 2 clinical trial, with 89% of patients reaching undetectable minimal residual disease after a median treatment duration of only 10 months [94]. Ultimately, due to the same mechanism of action, zanubrutinib treatment regimens are just as susceptible to C481S mutation of BTK as the other covalent inhibitors ibrutinib and acalabrutinib [100].

It is likely that the resistance mechanisms of all covalently binding BTK inhibitors share similar properties. Newly proposed, non-covalent BTK inhibitors might overcome the above-described issue of resistance mutations and serve as a subsequent therapy option for patients resistant to covalent inhibitors. Pirtobrutinib (LOXO-305) is a novel non-covalent inhibitor with a high selectivity for BTK described in an ongoing study of CLL patients previously treated with covalent BTK inhibitors [101]. Noticeably, pirtobrutinib inhibits both wild type and C481-mutated BTK and showed a better safety profile than ibrutinib, with fewer treatment-related adverse events and lower toxicity rates. Treatment with pirtobrutinib achieved an ORR of 62%, similar to CLL patients with covalent BTK inhibitor resistance and represents a promising alternative therapy choice to covalent inhibitors [101]. A summary of the properties of these BTK inhibitors is shown in Table 1.

Importantly, CLL can progress into a more aggressive lymphoma, known as Richter’s transformation, observed in about 5–10% of patients, with poor clinical outcome [102,103]. In patients under ibrutinib therapy, Richter’s transformation has an incidence rate of 3–8%, depending on the trial set-up and follow up times [104,105,106]. It would be useful to identify the risk of Richter’s transformation on treatment with any BTK inhibitor at the earliest possible time point, before the clinical appearance. In this respect, there is an ongoing discussion on the value of predictive biomarkers for therapy success and to predict side effects as outlined in the next chapter. 

## 6. Prognostic and Predictive Markers in the Era of BTK-Inhibitor Therapy

The era of BTK-inhibitor therapy raised the question of whether the established prognostic markers keep their power to predict the risk for progression under BTK inhibition. Prognostic markers in CLL can be divided into cell-intrinsic and extrinsic factors [107]. Intrinsic factors that drive disease progression include genomic alterations and mutations, such as the deletion of chromosome bands 11q, 17p (del11q, del17p) and trisomy of chromosome 12 (tri12), whereas del13q14.3 is indicative of a favorable disease prognosis [108,109]. Besides lack of the whole *TP53* gene by deletion of 17p, *TP53* mutations are associated with dismal prognosis and resistance to chemotherapy [78,110,111]. In relapsed or refractory patients harboring del17p, ibrutinib treatment leads to an overall response rate of 83%, with a 24-month PFS of 63% [104]. Patients with multiple *TP53* mutations have a significantly shorter overall survival and PFS compared to single-mutation cases under ibrutinib [104]. Furthermore, patients with del11p have significantly longer PFS of 70% over 42 months compared to 65% PFS in patients without del11p [106].

The mutational status of immunoglobulin heavy-chain variable region gene (IGHV) functions as a prognostic factor and is included in the international prognostic index for CLL. Leukemic cells with mutated and unmutated IGHV are thought to originate from different subsets of B cells [112]. Unmutated IGHV status is generally associated with dismal prognosis in CLL, with FCR (fludarabine, cyclophosphamide, rituximab) chemoimmunotherapy or bendamustine and rituximab-treated mutated IGHV patients showing treatment-free survival of 90% and 91% and unmutated IGHV patients showing 76% and 53% treatment-free survival [112]. Furthermore, FCR treatment resulted in significantly lower PFS in IGHV unmutated compared to mutated patients with a PFS rate of 33.1 versus 66.6% after five years. Mutational status did not impact overall response rates or minimal residual disease negativity in this study [113]. Mutated, fit IGHV patients with a generally favorable prognosis seem to benefit most from treatment with FCR [114]. Although IGHV mutational status can have an impact on the outcome of CLL patients under ibrutinib therapy (observation time five years) [7], especially high-risk unmutated IGHV patients benefit from ibrutinib treatment, as reported for combination treatment with rituximab [115,116]. Among the established biomarkers in CLL, CD49d is gaining more and more traction. Patients can be categorized in CD49d low (<30% positive CLL cells) and CD49d high (≥30%) cases based on the percentage of CD49d expression in CLL cells, with the CD49d high cohort having lower overall and treatment-free survival probability independent of the presence of other negative prognostic markers [117,118]. Even in the earliest stages of the disease, CD49d expression is a risk factor for treatment-free survival [119]. Furthermore, CD49d expression is commonly associated with high CD38 expression and unmutated IGHV status; however, CD49d high IGHV mutated cases have a worse prognosis than CD49d low IGHV mutated cases [120]. Together with short telomere length signifying a fast proliferative history of leukemic cells, CD49d expression concurs with increased genetic CLL clone instability and the combination of both prognostic markers identified patients with significantly shorter treatment-free survival as compared to patients with low CD49d expression and long telomeres [121]. It is not just CD49d expression, but also the expression pattern that is crucial in CLL disease prognosis, as bimodal expression of CD49d was shown to associate with a similarly worse prognosis than the previously described CD49d-high patients [122]. Notably, CD49d is most highly expressed in tri12 patients [123], which are characterized by high leukemic cell proliferation and a propensity towards Richter syndrome transformation [124]. In 89% of tri12 cases, a CD49d high status was found, likely due to hypomethylation of the CD49d gene ITGA4 [123]. Importantly, the prognostic value of CD49d expression over other flow cytometry-based markers such as CD38 and ZAP-70 is higher, as confirmed by a combined analysis of 2972 CLL cases, with CD49d high patients having significantly lower overall survival at 62% after 10 years of follow up, as compared to 84% in CD49d low patients [125]. Superiority in predicting overall survival was also found compared with other prognostic factors, namely mutational status of *TP53*, NOTCH1, SF3B1 and BIRC3 [126]. NOTCH1 mutations are known to be overrepresented in tri12 cases and NOTCH1 pathway engagement resulted in a positive CD49d expression regulation driven by NOTCH1-dependent activation of NF-κB these predictive markers together [127,128]. In the treatment context, high CD49d expression (≥30% positive CLL cells) identifies cases with reduced ibrutinib-induced lymphocytosis and lower nodal responses (Figure 2). Furthermore, CD49d high cases present with shorter PFS under ibrutinib-treatment [7,122]. The reasons might be found in the dual role of the integrin in shaping adhesion and BCR signaling.

## 7. Potential Bypass of BTK in the VLA-4 Signaling Activation Cascade

The VLA-4 ligand VCAM-1 is present on follicular dendritic cells, which simultaneously present an antigen that activates the BCR. VLA-4 is expressed in B cells and strengthens the antigen-BCR-synapse by being a part of it. In murine B cells the promoting effect of VLA-4 on the BCR-antigen-synapse was particularly potent if the B cell receptor affinity for the presented antigen was low [129]. In the context of CLL, the BCR-VLA-4-interaction is likely reciprocal; this means, VLA-4 has not only an effect on the BCR synapse but BCR-engagement conversely induces VLA-4 activation, resulting in conformational changes of the integrin, and thus, stronger tumor cell adhesion to follicular dendritic cells, creating positive feedback to the BCR. It is therefore logical that the CD49d low patient group with its characteristic absence of the integrin benefits more from BTK inhibition than the CD49d high group with its resistance loops of BCR initiation. CLL cells from ibrutinib-treated patients have a reduced adhesive capacity to fibronectin [130]. High concentrations of ibrutinib have also been described to decrease tumor cell adhesion in vitro [131]; however, when discussing BTK involvement in integrin inside-out activation, off-target kinase inhibition by this drug needs to be taken into account, and moreover, the reduction in baseline VLA-4 activation under ibrutinib can be overcome by antigen stimulation [7]. In particular, BCR-stimulated VLA-4 activation in CLL cells derived from patients that were 90 days on ibrutinib was comparable to the activation observed in untreated patients; this leads to the question of whether BTK can be bypassed by other kinases during antigen-stimulated inside-out VLA-4 activation. If this is the case, it could explain the decreased lymphocytosis and nodal response of CD49d high patients under ibrutinib. In these patients, BTK inhibition alone does not appear to be sufficient to disrupt the CLL residence in lymphoid organs, making patients more susceptible to developing resistance to therapy [7]. In vitro, combination treatment with both, ibrutinib and the PI3Kδ inhibitor idelalisib efficiently suppressed BCR-induced VLA-4 activation, pointing to PI3K being involved in the bypass cascade [7].

Current PI3K inhibitors used in CLL therapy include the PI3Kδ inhibitor idelalisib and its PI3Kγ/δ- and pan-inhibitor successors duvelisib and copanlisib, respectively [132]. The yet unapproved PI3Kδ inhibitor umbralisib was tested in combination studies with ibrutinib and the anti-CD20 antibody ublituximab, showing overall reduced adverse effects compared to other PI3K inhibitors [132,133,134]. Preliminary combination treatment of the novel PI3Kδ inhibitor zandelisib (MEI-401) with the BTK inhibitor zanubrutinib in relapsed or refractory CLL resulted in a complete response rate and was well tolerated, although only 5 patients were enrolled [94]. In a single-agent study using zandelisib with first continuous, then intermittent oral administration an ORR of 100% was reached in all three enrolled CLL patients under monotherapy and was likewise well tolerated [135]. The inhibitors described are without doubt active compounds, but there is further need to find the best way of administration in regard to patient comfort. 

Especially therapy discontinuation due to adverse events is an issue for PI3K-inhibitor treatment. In a phase 3 trial comparing combination therapy of idelalisib plus rituximab versus monotherapy with the BTK inhibitor acalabrutinib, 56% of patients under combination therapy suffered from severe adverse events compared to 11% in the acalabrutnib-treated cohort [136]. Therapy discontinuation was 47% in the idelalisib-rituximab-treated group, compared to 11% of patients treated with acalabrutinib. The authors discussed that this might have contributed to the significantly shorter PFS observed in the combination-treatment cohort [136]. Other combination therapies with idelalisib did not result in increased efficacy, but rather led to a higher frequency of serious adverse events and premature termination of two clinical trials [137,138,139]. Similar PI3K inhibitor-specific toxicities could be observed with the second-generation inhibitor duvelisib, where one study reported treatment discontinuation due to severe adverse events in 36% of duvelisib-treated patients [140]. Another downside of PI3K-targeted therapy are resistances, that for example can be caused by down-stream MAPK signaling pathways as reported by Murali et al. [141]. The authors demonstrated activating mutations in several components of MAPK signaling, such as MAP2K1, and that PI3K∂ inhibition failed to inhibit ERK phosphorylation in nonresponder CLL cells [141]. Furthermore, enhanced MAPK signaling was reported in a murine mouse model of PI3K∂ inhibitor resistance and in cells derived from idelalisib-treated patients with decreased sensitivity to the PI3K-inhibitor [142]; these reports suggest that MAPK-inhibitors might be interesting targets for treatment-resistant cases. 

Importantly, targeting the anti-apoptotic protein BCL-2 by the antagonistic agent venetoclax is effective in relapsed CLL patients, including those with negative prognostic features [143]; however, venetoclax resistance development can be observed in some cases, potentially mediated through the compensatory upregulation of other anti-apoptotic proteins from the BCL-2 family. Sensitivity to venetoclax might be restored through treatment with SYK and PI3Kδ inhibitors, emphasizing the importance of combination therapy with novel agents as shown in a clinical study demonstrating the efficacy of BTK inhibitor therapy in venetoclax-resistant CLL patients [144,145].

Murine models represent a powerful tool as they mimic the tumor microenvironment more faithfully than in vitro cultures. Recently, the therapeutic benefit of the PI3Kγ/δ inhibitor duvelisib in overcoming ibrutinib resistance was demonstrated using a murine xenograft model [146]. A widely used immunocompetent model for studying CLL is the Eµ-TCL1 transgenic mouse (TCL1 tg), resembling the IGHV unmutated CLL subgroup [147,148]. Similar to human CLL, an antigen-driven, CLL-dependent shift in T-cell subsets at various CLL stages is observed in these mice [149]; this model resembles the high-risk, CD49d-high CLL subgroup of patients, both in the primary TCL1 tg mouse model but also in transplantation settings [150]. IgM stimulation of splenic leukemic cells induces VLA-4 activation and cannot be overcome by BTK inhibition but by PI3Kd inhibition, analogous to the human high-risk situation [150].

A model to directly study BTK inhibitor resistance is the recently described C481S knock-in mouse, which was reportedly resistant to irreversible BTK inhibitors ibrutinib, acalabrutinib and zanubrutinib, but sensitive to the reversible inhibitor RN486 in vivo [151]. Another study developed and characterized Rag2-/-ɣc-/- xenograft mouse models transplanted with ibrutinib-resistant MEC-1 cells harboring either a BTKC481R or BTKC481S mutation; however, although the study showed ibrutinib-resistance of the generated BTK-mutated cell lines, there was no evaluation of drug-responses in the transplanted mouse model [152].

Overall, mouse models represent a useful tool in studying the efficacy of alternative therapy options on the background of resistance-mutations found in CLL patients progressing under therapy and can be utilized to further dissect the BCR downstream signaling by evaluating a potential BTK bypass mechanism in the VLA-4 activation cascade.

## 8. Conclusions

To date, BTK inhibitor therapy is a pillar in the treatment of CLL patients but toxicities or resistance development are still causing ibrutinib therapy discontinuation. Finding alternative treatment regimens with fewer off-targets and side effects aimed at combating therapy resistance in CLL is still an ongoing effort. To this end, understanding the mechanisms behind the occurrence of microenvironment-induced resistance will help to predict and overcome therapy resistance in the era of targeted therapy. 

## Figures and Tables

**Figure 1 cells-11-02235-f001:**
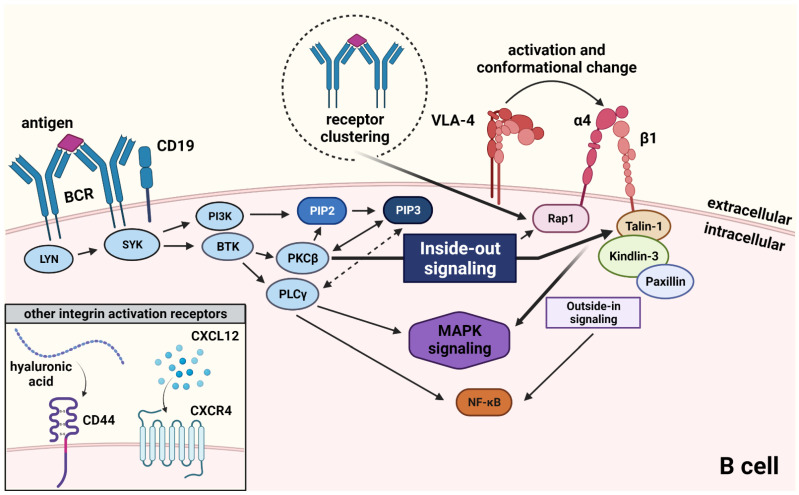
Inside-out signaling in antigen-mediated VLA-4 activation. Antigen-binding to the BCR initiates a downstream signaling cascade that involves several kinases and adaptor proteins and leads to the conformational change which is required for activation of VLA-4, a process called inside-out signaling. VLA-4 activation can also be initiated by other cues, as for example by binding of hyaluronic acid to CD44 or by binding of CXCL12 to CXCR4. Integrin activation induces, among others, MAPK signaling in a separate process, called outside-in activation.

**Figure 2 cells-11-02235-f002:**
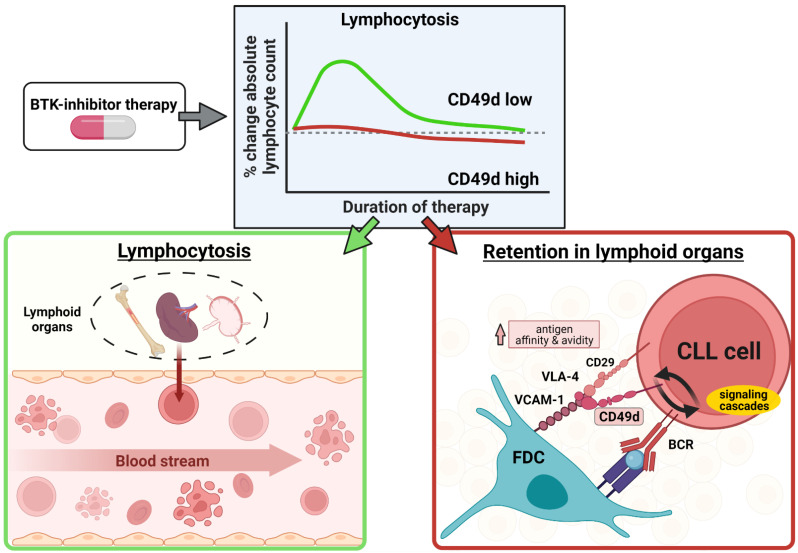
Possible mechanism of CD49d-mediated BTK-inhibitor resistance. CD49d high patients experience reduced lymphocytosis under BTK-inhibitor treatment as compared to CD49d low patients (**top**). In CD49d low cases (**left**) the mobilization of CLL cells out of lymphoid organs is induced upon BTK-inhibitor treatment start. Mobilized leukemic cells enter the bloodstream and succumb due to missing microenvironmental stimuli. In CD49d high cases (**right**), interaction of CLL cells with FDCs via VCAM-1 and BCR-mediated antigen-dependent signaling cascades lead to VLA-4 activation and retention in lymphoid organs.

**Table 1 cells-11-02235-t001:** Comparison of BTK inhibitors.

	Ibrutinib	Acalabrutinib	Zanubrutinib	Pirtobrutinib
**Mechanism of action**	Covalent	Covalent	Covalent	Non-covalent
**Approval**	02/2014, CLL	11/2019, CLL	11/2019, MCL	Phase I/II clinical trial (BRUIN)
**Most prevalent adverse events**	Infection Diarrhea Bleeding/Bruising	Infection Bleeding/Bruising Headache	Infection Neutropenia	Fatigue Diarrhea Contusion
**Most prevalent resistance mutations**	C481S/F/C/R, T474, L528, T316A, PLCγ2 mutations			Patient data not available yet

## Data Availability

Not applicable.

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
