# Peer review of "Integrin Signaling Shaping BTK-Inhibitor Resistance"

_cells, 2022, doi:10.3390/cells11142235_

Round 1

Reviewer 1 Report

In this interesting and comprehensive review, the authors describe in detail the role of the integrin VLA-4 in the pathogenesis of chronic lymphocytic leukemia. They describe how this integrin works and the inside-out signaling pathways controlling its activation. They also describe the class of Btk inhibitors, their success and their side effects, suggesting a direct inhibition of VLA-4 or of the activation pathway as an alternative strategy to overcome the problems of these therapies.

The work is well-structured and well-written. Two points that need to be addressed to improve the clarity of the manuscript are listed below.

1)      It appears clear that the inside-our signaling cascade is central in the manuscript. A figure or a scheme depicting in detail the signaling cascades implicated in this process might strongly improve the clarity of this section.

2)      The paragraph “Potential bypass of BTK in the VLA-4 signaling activation cascade” describes how VLA-4 signaling does not necessarily requires Btk for its activation, suggesting that CD49d might be implicated in resistance to Ibrutinib. However, I found that, although from the abstract it appears clear that the authors open to the possibility to develop new strategies to overcome the problems related to Btk inhibitors, they do not sufficiently discuss about it in the main text. I suggest that the authors expand the last part of the paragraph in this respect.

Author Response

We would like to thank Reviewer 1 for the insightful comments. We have implemented the suggestions as outlined in the point-by-point reply below.

Point 1:  In this interesting and comprehensive review, the authors describe in detail the role of the integrin VLA-4 in the pathogenesis of chronic lymphocytic leukemia. They describe how this integrin works and the inside-out signaling pathways controlling its activation. They also describe the class of Btk inhibitors, their success and their side effects, suggesting a direct inhibition of VLA-4 or of the activation pathway as an alternative strategy to overcome the problems of these therapies.

The work is well-structured and well-written. Two points that need to be addressed to improve the clarity of the manuscript are listed below.

Response 1: Thank you for the positive feedback.

Point 2: It appears clear that the inside-our signaling cascade is central in the manuscript. A figure or a scheme depicting in detail the signaling cascades implicated in this process might strongly improve the clarity of this section.

Response 2: We agree with this suggestion and have implemented an additional figure depicting the complex inside-out signaling cascade leading to VLA-4 activation..

Point 3: The paragraph “Potential bypass of BTK in the VLA-4 signaling activation cascade” describes how VLA-4 signaling does not necessarily requires Btk for its activation, suggesting that CD49d might be implicated in resistance to Ibrutinib. However, I found that, although from the abstract it appears clear that the authors open to the possibility to develop new strategies to overcome the problems related to Btk inhibitors, they do not sufficiently discuss about it in the main text. I suggest that the authors expand the last part of the paragraph in this respect.

Response 3: We agree with the suggestion and added an additional part into the last paragraph “Potential bypass of BTK in the VLA-4 signaling activation cascade” which discusses current available alternatives to BTK-inhibitor therapy, namely PI3K-, MAPK- and BCL-2 inhibitors, as well as current outcomes of combination studies utilizing BTK inhibitors with drugs targeting alternative components of the B cell receptor downstream pathways.

Reviewer 2 Report

B-cell receptor targeting agents are being used for the treatment of B-cell lymphoproliferative disorders, including chronic lymphocytic leukemia/CLL; indeed, Bruton’s tyrosine kinase/BTK-inhibitors represent an important therapeutic strategy for CLL patients, showing significant anti-tumor activity. However, BTK inhibition also presents with major challenges, such as indefinite use, side effects and development of drug resistance. Dr. Polcik and Collegues have provided a comprehensive review article, discussing: the inside-out signaling cascade leading to VLA-4 activation; pros and cons of BTK inhibition within the specific context of CLL; and, finally, refer to the potential mechanisms responsible for cell adhesion-mediated drug resistance.

Overall, this is an interesting review; it is well written and clearly presented.

Minor comment. Certain sections of the manuscript are very dense and it could be useful for the Reader to have a Table summarizing the main point. For instance, this could be applied to the different BTK inhibitors (ibrutinib, acalabrutinib, zanubrutinib, pirtobrutinib), by providing a table that support a direct comparison among them.

Author Response

We would like to thank Reviewer 2 for the insightful comments. We have implemented all suggestions as outlined below.

Point 1:  B-cell receptor targeting agents are being used for the treatment of B-cell lymphoproliferative disorders, including chronic lymphocytic leukemia/CLL; indeed, Bruton’s tyrosine kinase/BTK-inhibitors represent an important therapeutic strategy for CLL patients, showing significant anti-tumor activity. However, BTK inhibition also presents with major challenges, such as indefinite use, side effects and development of drug resistance. Dr. Polcik and Collegues have provided a comprehensive review article, discussing: the inside-out signaling cascade leading to VLA-4 activation; pros and cons of BTK inhibition within the specific context of CLL; and, finally, refer to the potential mechanisms responsible for cell adhesion-mediated drug resistance.

Overall, this is an interesting review; it is well written and clearly presented.

Response 1: Thank you for the positive feedback.

Point 2:  Minor comment. Certain sections of the manuscript are very dense and it could be useful for the Reader to have a Table summarizing the main point. For instance, this could be applied to the different BTK inhibitors (ibrutinib, acalabrutinib, zanubrutinib, pirtobrutinib), by providing a table that support a direct comparison among them.

Response 2: We gladly followed this suggestion. We implemented an additional figure depicting a simplified version of the integrin-activating inside-out signaling cascade. We also added an overview of current BTK inhibitors to the chapter “Next-generation BTK inhibitors”, including their mechanism of action, approval dates, most common resistance mutations and the most prevalent adverse events.